# OpenReview forum: "Anchored Decoding: Provably Reducing Copyright Risk for Any Language Model"
_ICML.cc/2026/Conference — ICML 2026 regular_

### Official Review · Reviewer_tgex · 2026-03-03

**Soundness:** 4
**Presentation:** 3
**Significance:** 3
**Originality:** 3
**Overall Recommendation:** 4
**Confidence:** 2

**Summary:**

This paper proposes Proximal Decoding: an inference-time method that mitigates LM’s verbatim reproduction of copyrighted text.

The core idea is to fuse the next-token distributions of a "risky" LM (potentially trained on copyrighted content) and a "safe" LM (trained only on public domain data) at each decoding step, subject to a per-step KL divergence budget relative to the safe model.

Two additional mechanisms improve the approach: a prefix debt that detects memorization-prone prompts via log-likelihood ratio spikes and preemptively tightens the budget, and an adaptive banking rule that rolls unused budget forward.

To handle tokenizer mismatches between safe and risky models, the authors introduce ProximalByte Decoding, a byte-level variant leveraging the ByteSampler framework. They also contribute with Comma 1.7B, a model trained on openly licensed data with the Llama 3.1 tokenizer.

Across multiple model pairs, Proximal Decoding consistently achieves the best balance between reducing copying and preserving output quality compared to existing baselines, eliminating up to 75% of the measurable copying gap between the risky and safe models, while keeping fluency and factuality close to the risky model's level. The added inference cost is modest (roughly 1.1× slower).

**Compliance With Llm Reviewing Policy:**

Affirmed.

**Final Justification:**

I'm satisfied with the authors' response, so my evaluation of the paper remains positive.

**Key Questions For Authors:**

- Regarding W2: if a user pastes a copyrighted chapter into the prompt and asks for a summary, would the prefix debt mechanism interpret the high LLR spikes as a memorization trigger? If so, how could a practitioner distinguish between "the model memorized this" and "the user provided this"?
- The paper evaluates only base models. Instruction-tuned models may have different memorization profiles. Do you have any indication of whether the method transfers to that setting?
- How sensitive is the method to the quality gap between $p_s$ and $p_r$? If a stronger safe model were available (e.g., closer in capability to the risky model), would you expect the utility cost to shrink significantly?

**Limitations:**

yes

**Strengths And Weaknesses:**

**Strengths**

- S1. The paper addresses an increasingly growing problem with a solution grounded in information theory. The budget parameter $K$ gives users a clear, interpretable way to control the balance between safety and output quality.
- S2. The experimental design is thorough: six model pairs, three risky models, two safe models, evaluated at both token and byte levels. The ablation study in Section 5.2 systematically validates each design choice and shows that each component contributes.
- S3. Beyond the algorithm, the paper will contribute with the weights of Comma 1.7B, a language model trained exclusively on permissively licensed data, which could be valuable for other researchers working on copyright-safe applications.

**Weaknesses**

- W1. The copyright evaluation relies only on 16 novels. This is a relatively small and genre-homogeneous sample. While the authors claim that "Proximal Decoding is agnostic to tokenization, modality, and domain," the paper does not evaluate copyright risk on other high-risk domains such as news articles, where writing style and content structure may differ significantly. Therefore, it is unclear whether the method generalizes beyond literary works or whether it is particularly suited to long-form fiction.
- W2. Proximal Decoding is applied at every decoding step regardless of whether copyright risk is actually present. It is unclear how the method affects general-purpose quality on tasks like summarization, question answering. Additionally, the method is designed to suppress reproduction of memorized training data, but does not address scenarios where copyrighted content is provided by the user in the prompt (e.g., asking the model to summarize a copyrighted document).
- W3. While aggregate metrics are reported effectively, there is little instance-level analysis of when the method fails. When verbatim is reproduced, what does that look like?

---

> ### Author Rebuttal · Authors · 2026-03-30
>
> We thank reviewer **tgex** for their time and feedback! Responding point-by-point:
>
> **“[W1] The paper does not evaluate copyright risk on other high-risk domains such as news articles…it is unclear whether the method generalizes beyond literary works.”**
>
> We added a new evaluation on a high-risk news domain—following [7], we consider 560 NYTimes articles from NewsSpan with the model pair (Comma 1.7B, Llama 3.1 70B). For copyright risk, we report NCR as in the paper. For utility, following CotaEval [6], we use a reference-free pooled pairwise win rate, where an LLM judge compares each method’s continuation against outputs from a fixed competitor pool on randomly sampled prompts.
>
> Our [trade-off plot for this domain](https://ibb.co/Z1gfwxV2) shows that Proximal Decoding exhibits a similarly favorable risk-utility tradeoff, suggesting good generalization beyond literary works.
>
> **“[W2] It is unclear how [Proximal Decoding] affects general-purpose quality on tasks like summarization, question answering.”**
>
> As a lightweight sanity check, we take the model pair (Comma 1.7B, Llama 3.1 70B) and evaluate Proximal Decoding at k=1.5, one evaluated operating point in the high-protection region (Fig. 1b), on downstream factual QA (TruthfulQA MC1/MC2), summarization (CNN/DailyMail), and code generation (HumanEval pass@1), using lm-eval on 400 samples per task.
>
> | Model | TruthfulQA–MC1 Acc. | TruthfulQA–MC2 Acc. | CNN/DailyMail ROUGE | HumanEval pass@1 |
> |---|---:|---:|---:|---:|
> | $p_s =$ Comma 1.7B | 0.255 | 0.431 | 0.092 | 0.018 |
> | ProxDec ($k=1.5$) | **0.303** | 0.472 | 0.139 | 0.488 |
> | $p_r =$ Llama 3.1 70B | 0.300 | **0.473** | **0.156** | **0.506** |
>
> At a practically useful operating point, Proximal Decoding substantially improves over $p_s$ and remains close to $p_r$ across each task.
>
> **“[Q1] [Proximal Decoding] does not address scenarios where copyrighted content is provided by the user in the prompt...if a user pastes a copyrighted chapter into the prompt and asks for a summary, would the prefix debt mechanism interpret the high LLR spikes as a memorization trigger? How could [one] distinguish between "the model memorized this" and "the user provided this"?"**
>
> We agree; this scenario lies outside the primary threat model of our paper. Proximal Decoding is designed to mitigate copying attributable to parametric memorization of training data, not copyrighted text explicitly supplied in-prompt.
>
> The prefix debt is very large if the user-provided input is much more likely under the risky model than the safe model, which pushes early generation toward the safe model. This may reduce closeness to the source text, but also degrade summarization quality if the safe model is less capable. Crucially, prefix debt is not a provenance detector: it cannot distinguish overlap arising from parametric memorization versus user-supplied copyrighted text. Rather, it operationalizes the hypothesis that a sequence much more likely under $p_r$ than $p_s$ indicates asymmetric memorization risk, not definitive evidence of legal status. In deployments where user-supplied source text is authorized, disabling prefix debt is a straightforward remedy.
>
> **“[W3] There is little instance-level analysis of when the method fails. When verbatim is reproduced, what does that look like?”**
>
> We show qualitative examples in App. D.6. At higher KL budgets, verbatim reproduction typically appears as a contiguous copied span from the start before departing from the source text.
>
> We also systematically analyze this in App. E: verbatim copying is preceded by high per-step KL divergence between $p_r$ and $p_s$ (induced when the risky model is likely to have memorized the sequence, but the safe model hasn’t). Further, copying risk is concentrated in early generation (Fig. 9), which motivates the prefix debt.
>
> **“[Q2] Do you have any indication of whether the method transfers to [instruction-tuned LMs]?”**
>
> There’s no inherent obstacle to applying our purely inference-time method to instruction-tuned models. We focus on base models as our safe models lack instruction-tuned counterparts. To directly address this question, we ran a new experiment evaluating the mixed pair (Llama 3.1 70B Instruct, Comma 1.7B) and observe the same beneficial risk–utility tradeoff for Proximal Decoding [here](https://ibb.co/svCTqwV9).
>
> **“[Q3] How sensitive is the method to the quality gap between  $p_r$ and $p_s$? If a stronger safe model were available, [would] utility cost…shrink significantly?”**
>
> Yes. We evaluate safe models at different scales and empirically find that a stronger safe model reduces the utility cost of Proximal Decoding. Comparing Comma 1.7B vs. Comma 7B in Fig. 2, the stronger Comma 7B yields a better risk-utility frontier. This is consistent with the method's mechanism: as decoding is constrained to remain close to $p_s$, a higher-utility safe model imposes less distortion on the risky model’s preferred distribution at a fixed budget.

---

> > ### Author Rebuttal · Reviewer_tgex · 2026-04-04
> >
> > I thank the authors for their detailed response and for incorporating additional experiments during the rebuttal. My evaluation of the paper remains positive, and I will keep my original score.

---

> > > ### Author Response · Authors · 2026-04-04
> > >
> > > Thank you for your feedback and time! We are glad your evaluation of our paper remains positive.

---

### Official Review · Reviewer_JTGw · 2026-03-12

**Soundness:** 3
**Presentation:** 3
**Significance:** 3
**Originality:** 3
**Overall Recommendation:** 4
**Confidence:** 3

**Summary:**

This paper proposes PROXIMAL DECODING, an inference-time method that constrains the output of a copyright-risky language model to remain close to a permissively trained safe model. At each decoding step, the method fuses the next-token distributions of a risky model $p_r$ and a safe model $p_s$ while limiting deviation from the safe model under a user-specified budget $K$. The paper formalizes this through the K-NAF (K-Near Access-Freeness) framework, approximating the sequence-level KL-constrained objective with per-step constrained optimization and arguing that the local constraints compose into a global guarantee. The method also introduces prefix debt, which reduces the initial budget when the prefix shows strong memorization signals, and adaptive banking, which reallocates unused budget across decoding steps.
To support models with different tokenizers, the paper also introduces PROXIMALByte DECODING, which lifts both models to byte-level next-byte distributions using ByteSampler.

Experiments evaluate copyright risk on BOOKS and utility on BIOS. Risk is measured with NCR (normalized copying reduction), which aggregates six overlap-based metrics over copyrighted text snippets, while utility is measured via continuation fluency and factuality (FActScore). Across six model pairs, PROXIMAL and PROXIMALByte generally achieve favorable risk–utility trade-offs and form the Pareto frontier in the high-protection regime (NCR $\ge 75\%$). Ablation studies further suggest that the optimization objective, prefix debt, and adaptive budgeting contribute to improved performance, with a reported runtime overhead of about 1.1× for tokenizer-matched decoding.

**Compliance With Llm Reviewing Policy:**

Affirmed.

**Final Justification:**

I am satisfied with the authors’ response, and my overall assessment of the paper remains positive.

**Key Questions For Authors:**

Q1. NCR is ultimately a relative metric defined with respect to a risky baseline and a safe reference model. How well does this metric track actual copyright risk reduction in practice? In particular, if the safe model $p_s$ contains latent leakage, is NCR-based evaluation still a reliable proxy?

Q2. The utility evaluation is split between BOOKS fluency and BIOS factuality. Could the authors add an evaluation that directly measures copying suppression and semantic usefulness on the same task or input space? At present, the utility preservation story remains somewhat indirect.

**Limitations:**

yes

**Strengths And Weaknesses:**

S1. The problem setting is timely and well-motivated.
The paper addresses a practical gap: retraining a fully copyright-clean model is expensive, yet deploying a risky model as-is carries potential regurgitation concerns. The idea of using a safe model as a reference distribution and enforcing post-hoc control at inference time is appealing from a deployment perspective. The fact that the method is training-free and requires only off-the-shelf model logits further strengthens its practical relevance.

S2. The theoretical development is clean and reasonably convincing.
The paper starts from a sequence-level constrained objective, relaxes it into per-step KL-constrained optimization, and then uses compositional arguments to derive a global K-NAF guarantee. This is conceptually simple, but effective. The fact that the solution admits a closed-form weighted geometric mean also helps the method feel principled rather than heuristic. The additions of prefix debt and adaptive banking are also intuitive, reflecting the observation that not all decoding steps are equally risky.

S3. The byte-level extension is a meaningful practical contribution.
Many two-model fusion methods implicitly assume a shared vocabulary, which is a severe limitation in realistic deployment settings. In contrast, PROXIMALByte DECODING uses byte-level distributions to bypass tokenizer mismatch. This substantially broadens the method’s applicability: in the experimental setup, only one out of six model pairs is tokenizer-matched, so this extension is not merely cosmetic but materially expands coverage.

S4. The empirical message is fairly consistent.
The focus on the high-protection regime, i.e., NCR $\ge 75\%$, is well aligned with the intended application. In Table 1 and Figures 1–2, the PROXIMAL variants show favorable risk-utility trade-offs relative to the baselines, and the ablation studies indicate that prefix debt and adaptive budgeting contribute meaningfully. The efficiency results are also encouraging: compared with other joint-model decoding baselines, the runtime overhead is modest.

---
W1. My main concern is the assumption that the safe model provides an adequate gold-standard notion of safety.
NCR is fundamentally a relative metric: it measures how much the risky model’s behavior is shifted toward that of the safe reference model. As a result, the method does not directly guarantee that copyright infringement risk is minimized in any legal or absolute sense; rather, it guarantees bounded proximity to a chosen anchor model. This leads to a more limited interpretation of the claimed guarantee.

W2. While the claim of 'provably reducing copyright risk' aligns with the observed direction, it may slightly overstate the current findings.
What is provable is bounded deviation from $p_s$ under the K-NAF framework, not direct reduction of real-world legal or infringement risk. The authors also explicitly note that the local per-step optimization is only an approximation to the global sequence-level constrained optimum. Thus, the guarantee is best understood as a formal bound within a proxy safety framework, rather than a direct legal or practical safety guarantee. The title and early framing are somewhat stronger than the exact scope of the formal guarantee.

W3. The evaluation protocol remains fairly proxy-heavy.
Copyright risk is assessed entirely through overlap-based metrics, while utility is split across BOOKS fluency and BIOS factuality. This means that copying suppression and semantic usefulness are not evaluated on the same task or input distribution. Moreover, the fluency score depends on a model-based judge, which introduces additional evaluator dependence. The protocol is not unreasonable, but it does not fully support the strength of the paper’s broader copyright-risk claims.

---

> ### Author Rebuttal · Authors · 2026-03-30
>
> We thank reviewer **JTGw** for their time and feedback. Responding point-by-point:
>
> **“[W1] NCR is fundamentally a relative metric…As a result, the method does not directly guarantee that copyright infringement risk is minimized in any legal or absolute sense…[Q1] How well does [NCR] track actual copyright risk reduction in practice? If the safe model  contains latent leakage, is NCR-based evaluation still a reliable proxy?”**
>
> Both our theoretical guarantee and empirical NCR metric are indeed relative, not absolute, notions of risk. The K-NAF result guarantees that the decoding stays within a user-controlled divergence budget of the safe model; it does not certify minimal copyright risk in any legal or model-independent sense. Likewise, NCR is not part of the guarantee, nor a legal measure of infringement risk, but a comparative evaluation metric that aggregates 6 copying indicators (e.g., ROUGE, LCS, ACS) to measure how much copying is reduced without collapsing generation quality. Tables 15–20 show the absolute breakdown of underlying copying metrics, from which we derive Figs. 1, 2, and Table 1.
>
> The key assumption is that the safe model is meaningfully lower-risk than the risky model, which is supported in our experiments both by construction and empirically: our safe models have openly documented training-data provenance that excludes the copyrighted books used in our evaluation, and exhibit low absolute copying values (Tables 15-20).
>
> Thus, while neither K-NAF nor NCR provides an absolute legal guarantee, they do ensure a meaningful relative guarantee and proxy evaluation under the explicit assumption that the safe model is a genuinely copyright-free reference. We’ll clarify this in revision.
>
> **“[W2] The claim of provably reducing copyright risk...overstates current findings. What is provable is…under the K-NAF framework, not direct reduction of real-world legal or infringement risk.”**
>
> We agree that the formal scope should be stated precisely. Our K-NAF result is a guarantee that the decoded distribution provably remains within a controlled divergence budget of a permissively trained model, not a direct legal certification of non-infringement. Please see our response to **zKjq**, where we discuss the distinction between the K-NAF guarantee and legal copyright risk. We’ll revise the framing accordingly.
>
> **“[W2] The local per-step optimization is only an approximation to the global sequence-level constrained optimum. Thus, the guarantee is best understood as a formal bound within a proxy safety framework, rather than a direct legal or practical safety guarantee.”**
>
> More precisely, the “approximation” here concerns the optimization objective—i.e., the relationship between the local decoding objective (Eq. 4)  and the original sequence-level constrained optimization problem (Eq. 3)—not the validity of the K-NAF guarantee itself. Our method yields a feasible, though not necessarily globally optimal, solution. Thm. 3.1 shows that the original guarantee exactly holds for the decoding rule used in the paper: the per-step constraints compose to satisfy the sequence-level bound.
>
> **“[W3] Copying suppression and semantic usefulness are not evaluated on the same task or input distribution. [Q2] Could the authors add an evaluation that directly measures copying suppression and semantic usefulness on the same task or input space?”**
>
> To clarify our evaluation setup, our 6 copying-suppression metrics on BOOKS are computed on the same generated outputs used for our fluency metric, so our evaluation directly measures copying risk and this form of utility on the same task and input space. Our factuality metric is evaluated separately on BIOS, which is indeed a different task setting. More broadly, such task split evaluation is common in copyright-evaluation work [5,6,7].
> We also conduct [new trade-off experiments](https://ibb.co/Z1gfwxV2) on a news domain where copyright risk and utility are evaluated on the same task space. We measure semantic usefulness using a new pairwise winrate metric. Proximal Decoding remains Pareto-optimal over pointwise baselines in this new evaluation; see our response to **tgex** for details.
>
> **“[W3] The fluency score depends on a model-based judge, which introduces additional evaluator dependence.”**
>
> LLM-as-a-judge does introduce evaluator dependence. However, this is common in related work [5,6,7]. All methods are scored using the same judge and rubric, so any systematic bias is shared across methods and is therefore less likely to confound relative comparisons. Note that we follow CopyBench exactly in our fluency evaluation setup.
>
> **Citations:**
>
> [5] Chen et al. CopyBench: Measuring Literal and Non-Literal Reproduction of Copyright-Protected Text in Language Model Generation. EMNLP 2024.
>
> [6] Wei et al. Evaluating Copyright Takedown Methods for Language Models. NeurIPS 2024.
>
> [7] Zhang et al. Certified Mitigation of Worst-Case LLM Copyright Infringement. EMNLP 2025.

---

> > ### Author Rebuttal · Reviewer_JTGw · 2026-04-03
> >
> > Thanks for the detailed response, especially the clarification on the K-NAF guarantee and the additional experiments. My concerns are resolved, and I am maintaining my score, which reflects my positive assessment of the paper.

---

> > > ### Author Response · Authors · 2026-04-03
> > >
> > > Thank you very much for your thoughtful feedback and positive assessment of our paper! We are glad that the rebuttal has addressed your concerns, and we sincerely appreciate your support.

---

### Official Review · Reviewer_zKjq · 2026-03-13

**Soundness:** 2
**Presentation:** 3
**Significance:** 2
**Originality:** 2
**Overall Recommendation:** 4
**Confidence:** 2

**Summary:**

This paper studies how to reduce the risk that a language model reproduces copyrighted training text without retraining the model. The setting assumes access to two models. A high-capability model that may memorize copyrighted data and a permissively trained model that is assumed to be safe because it was trained only on public or licensed data. They propose Proximal Decoding, a decoding strategy that combines these two models during generation, and use two additional mechanisms improve robustness.

**Compliance With Llm Reviewing Policy:**

Affirmed.

**Key Questions For Authors:**

1. How does the trade-off between quality and copyright risk change when the safe model has substantially lower capability than the risky model?
2. While the overhead is moderate in reported experiments, the cost may grow with larger safe models or deployment settings, right?

**Limitations:**

Please see the weakness

**Strengths And Weaknesses:**

Strength:
The problem formulation is clear and extensions for practical deployment is well-motivated.
Weakness:
1. The paper claims provable reduction of copyright risk. However, the theoretical guarantee only ensures that the generated distribution remains close to the safe model distribution. This guarantee does not directly imply bounds on memorization or copyright copying behavior. The argument then depends on the assumption that the safe model distribution represents a genuinely safe reference.
2. The optimization problem is defined at the sequence level, but the algorithm enforces constraints independently at each decoding step. The paper does not quantify how large this approximation gap may be.
3.The paper does not provide analysis showing how the quality gap between $p_r$ and $p_s$ affects the achievable utility under the KL constraint.

---

> ### Author Rebuttal · Authors · 2026-03-30
>
> We thank reviewer **zKjq** for their time and thoughtful feedback. Responding point-by-point:
>
> **“The theoretical guarantee [of provable reduction of copyright risk] only ensures that the generated distribution remains close to the safe model distribution, [which] does not directly imply bounds on memorization or copyright copying behavior. The argument then depends on the assumption that the safe model distribution represents a genuinely safe reference.”**
>
> Thanks for the observation. This is correct in spirit: our theoretical guarantee is relative to the safe reference model, not an absolute legal guarantee of non-infringement. More precisely, our theoretical result establishes a formal K-NAF [3] guarantee that the decoded distribution remains within a controlled KL budget of the safe model. By itself, this doesn’t imply a model-independent bound on memorization or copyright risk; its practical interpretation depends on the safe model being a genuinely safe reference.
>
> In our setting, we believe that is a reasonable assumption: the safe models are constructed from disclosed permissive data sources and exclude the copyrighted books used in evaluation, and empirically they show low copying values. We’ll revise the wording to make this dependence explicit.
>
> **“The optimization problem is defined at the sequence level, but the algorithm enforces constraints independently at each decoding step. The paper does not quantify how large this approximation gap may be.”**
>
> To clarify, while our decoding rule is a token-level approximation (Eq. 4) to the sequence-level optimization (Eq. 3), the global constraint is still satisfied: Theorem 3.1 shows that our local per-step procedure always produces a sequence distribution that is feasible for the sequence-level KL constraint—i.e., the risk bound holds globally—but we do not claim it is the global optimum. More broadly, this local-decoding relaxation is standard for related controlled generation methods [2,4].
>
> The approximation gap is therefore only with respect to the optimal objective value, not to satisfying the global constraint itself. We can’t quantify the exact approximation gap as enforcing this constraint at token $t$ requires knowing the risk contributions of all future tokens $t+1,...,T$, which are not available at decoding time without exhaustive lookahead. We’ll clarify this in revision.
>
> **“The paper does not provide analysis showing how the quality gap between  $p_r$ and $ p_s$ affects the achievable utility under the KL constraint.”**
>
> Thanks for pointing this out. We do provide empirical evidence for this dependence in Fig. 2. When holding the risky model fixed (e.g., for Qwen 2.5 72B or Llama 4 17Bx16E), replacing Comma 1.7B with a stronger safe model Comma 7B yields a more favorable frontier, with higher fluency/factuality at comparable levels of copying reduction. At the same time, the gap is not so large that utility collapses: even with the smaller Comma 1.7B, the achievable frontier remains strong. Thus, our results suggest that a larger quality gap does worsen the tradeoff, but in the regimes we study it does not make the method ineffective. This complements the intuition that when $p_s$ is closer to $p_r$, the KL constraint becomes cheaper to enforce.
>
> **“How does the trade-off between quality and copyright risk change when the safe model has substantially lower capability than the risky model?”**
>
> Even when the safe model is much weaker than the risky model, our results show that Proximal Decoding still achieves a strong risk-utility tradeoff. Our main experiments operate exactly in this regime: a much smaller safe model (Comma 1.7B) paired with much stronger risky models (>70B params) still preserves strong utility while substantially reducing copying risk.
>
> As the capability gap widens, the tradeoff does become less favorable: because the two models’ next-token distributions are farther apart, enforcing proximity to the safe model necessitates larger corrections to the risky model’s preferred distribution. We mitigate this in practice by 1. pre-training and releasing our own safe model (Comma 1.7B), which is the most performant safe model of its size (Fig. 4), and 2. introducing byte-level integration, which allows our method to leverage stronger safe models when available.
>
> **“While the overhead is moderate in reported experiments, the cost may grow with larger safe models or deployment settings, right?”**
>
> Larger safe models indeed raise the computational overhead. That said, our results show  that strong risk-utility tradeoffs are already achievable with much smaller safe models. While stronger safe models can improve the frontier further, our experiments suggest that matching the risky model’s scale is not necessary for good performance.
>
> **Citations:**
>
> [3] Vyas et al. On Provable Copyright Protection for Generative Models. ICML 2023.
>
> [4] Abad et al. Copyright-Protected Language Generation via Adaptive Model Fusion. ICLR 2025.

---

> > ### Author Rebuttal · Reviewer_zKjq · 2026-04-06
> >
> > i don't have further questions

---

### Official Review · Reviewer_tXTj · 2026-03-13

**Soundness:** 3
**Presentation:** 3
**Significance:** 3
**Originality:** 3
**Overall Recommendation:** 4
**Confidence:** 4

**Summary:**

This paper introduces "Proximal Decoding," an inference-time technique designed to prevent large language models (trained on mixed or risky data) from verbatim memorization and reproduction of copyrighted text. The core idea is to run a "safe" language model (trained exclusively on permissive data) in parallel with the risky model during generation. The method constrains the output token distribution of the risky model to stay within a specified KL-divergence radius of the safe model's distribution.

Beyond the empirical methodology, the authors provide theoretical grounding using PAC-Bayesian bounds to formally bound the probability of emitting copyrighted strings. They back this up with empirical evaluations (on models like LLaMA and Mistral) to show that Proximal Decoding effectively reduces memorization while mostly maintaining the model's core utility.

**Compliance With Llm Reviewing Policy:**

Affirmed.

**Key Questions For Authors:**

1. Your approach depends heavily on the quality of the safe model. Have you experimented with safe models of drastically different scales or qualities? How sensitive is the utility of the proximal decoded output to the safe model's perplexity?
2. Running two models at inference time severely impacts latency and throughput. Have you considered or tested any speculative or approximation techniques to reduce this overhead?
3. The constraint parameter seems to be applied uniformly. Is there any exploration into dynamically adjusting the KL bound per-token based on the entropy or confidence of the safe model?
4. How does this method behave on highly structured generation tasks like code generation (I know it might be a bit off of your goal) where the safe model might lack the syntax knowledge that the risky model possesses? Does the projection break the syntax?

**Limitations:**

The authors discuss theoretical limitations, but practical limitations are a bit lacking specially for an idea with such potential for adoption across the industry. I suggest squeezing things a bit and adding a dedicated paragraph addressing the severe inference latency/compute overhead of running dual models, and the practical difficulty of sourcing a highly capable "safe" model that doesn't drag down the risky model's performance on niche tasks.

**Strengths And Weaknesses:**

**Strengths:**
* **Soundness:** Framing the copyright/memorization issue as a constrained optimization problem at inference time is mathematically elegant. The theoretical contribution, specifically deriving the PAC-Bayesian bounds for the probability of generating copyrighted sequences, is rigorous and gives a solid backbone to the empirical claims. The experiments successfully demonstrate the tradeoff between utility and memorization.
* **Significance:** This addresses a massive, real-world pain point for generative AI right now. Finding ways to utilize the capabilities of models trained on wild web data while mathematically bounding copyright risk is highly relevant to the ICML community and the industry at large.
* **Originality:** While dual-model inference techniques (like contrastive decoding or speculative decoding) are well known, using a safe/risky model pair coupled with a KL-divergence constraint specifically for copyright mitigation is a fresh application. The theoretical bounds elevate the originality significantly.

**Weaknesses:**
* **Soundness / Assumptions:** The entire method relies heavily on the availability and capability of a "safe" model. In practice, models trained purely on permissive data are often significantly weaker or have skewed distributions compared to state-of-the-art mixed-data models. If the safe model's distribution is too flat or ignorant of a specific domain, forcing the risky model to stay close to it might aggressively degrade utility in ways the current benchmarks don't fully capture.
* **Presentation:** The paper is mostly clear, but the discussion around the computational overhead of running two models simultaneously needs more transparency. Inference is already a bottleneck for LLMs; doubling the memory and compute requirements is a massive practical hurdle that feels slightly glossed over in the narrative.
* **Originality (Context):** The authors could do a slightly better job differentiating the mechanics of their projection from standard contrastive decoding literature. While the end goal (copyright) is different, the structural inference-time mechanics are quite similar to existing alignment-via-decoding methods.

---

> ### Author Rebuttal · Authors · 2026-03-30
>
> We thank reviewer **tXTj** for their time and feedback! Responding point-by-point:
>
> **"The method relies heavily on the availability and capability of [safe] models [which] are often significantly weaker or have skewed distributions. Have you experimented with safe models of drastically different scales or qualities? How sensitive is the utility of the proximal decoded output to the safe model's perplexity?"**
>
> Safe models indeed are often weaker and trained on limited distributions, but our results show that Proximal Decoding works well in exactly that regime. In main experiments, the safe anchor is much smaller than the risky model (e.g., Comma 1.7B vs. Llama 3.1 70B), but still yields strong utility while substantially reducing copying risk. We also evaluate a larger safe model (Comma 7B) with stronger downstream performance (Fig. 4), confirming that safe-model quality matters quantitatively but is not a prerequisite for the method to be effective.
>
> More broadly, as Proximal Decoding constrains the decoded distribution to remain close to the safe anchor, weaker or higher-perplexity safe models do incur a larger utility cost at a fixed budget. This worsens the achievable risk-utility frontier, but does not invalidate the method entirely. We will revise the paper to make this point more explicit.
>
> **“I suggest addressing the severe inference latency/compute overhead of running dual models…Have you considered…speculative or approximation techniques to reduce overhead?”**
>
> We agree this deserves emphasis; as we’ve noted in Sec. 5.3, Proximal Decoding doesn’t inherently require a near-2x increase in compute or memory: in our experiments, the safe model is substantially smaller than the risky model  (e.g., 1.7B vs. 70B), so the overhead is closer to one large-model pass plus one small-model pass.  Accordingly, the slowdown is much less than 2x; for (Llama 3.1 70B, Comma 1.7B), throughput slows by only 1.1x  (Table 2), consistent with our FLOPs analysis (143.4 vs. 140 GFLOPs/token).
>
> To reduce overhead, our implementation incorporates several numerical/systems-level optimizations (App. B.4), including Newton-Raphson solving, prefill-caching, and a logit-gather trick. Standard speculative decoding [1] is difficult to efficiently apply here because the fused target distribution requires both model forward passes at each step; approximate variants would forfeit the guarantee, while exact variants would offer limited practical speedup.
>
> **“The constraint parameter seems to be applied uniformly...any exploration into dynamically adjusting the KL bound per-token based on the entropy/confidence of the safe model?”**
>
> Our per-step budget $k_t$ is already dynamic rather than uniform across tokens (Eq. 7): it depends on remaining budget and prior budget use, reflecting that copyright risk is distributed non-uniformly across tokens (App. E.3), and is defined to satisfy K-NAF (Prop. 3.4).  We do not currently tie $k_t$ to the safe model’s entropy/confidence;  we leave this promising extension to future work. We ablate alternative dynamic budget allocations in Sec. 5.2.
>
> **“How does this method behave on highly structured generation tasks like code generation…where the safe model might lack the syntax knowledge that the risky model possesses? Does the projection break the syntax?”**
>
> We agree this is an important limitation to clarify. If the safe model is substantially weaker on a structured generation task, Proximal Decoding may degrade performance because the feasible fused distribution is constrained to stay near the safe model’s distribution. This effect is more pronounced at tighter budgets. Thus, syntax preservation is not guaranteed when the safe model lacks sufficient in-domain coverage.
>
> As a lightweight sanity check, we evaluate the pair (Comma 1.7B, Llama 3.1 70B) on HumanEval, a code generation task. Proximal Decoding at $k=1.5$, a representative Pareto-efficient operating point (Fig. 1b), is close to Llama 3.1 70B alone (0.488 vs. 0.506 pass@1), suggesting that structured generation remains intact with a suitable model pair. See our response to **tgex** for details.
>
> **“The authors could [better] differentiate the mechanics of their projection from standard contrastive decoding.”**
>
> We agree this distinction should be made clearer. While Proximal Decoding uses a dual-model structure similar to contrastive decoding (CD) [2], the projection mechanism is different. In CD, the weaker model acts as a repulsive term, and decoding favors tokens with high expert-over-amateur scores to improve utility. In contrast, our projection is derived from an objective that explicitly trades off utility against copying risk. This distinction also underlies the theoretical guarantees for our method, which CD does not provide.
>
> **Citations:**
>
> [1]  Leviathan et al. Fast Inference from Transformers via Speculative Decoding. ICML 2023.
>
> [2] Li et al. Contrastive Decoding: Open-ended Text Generation as Optimization. ACL 2023.

---

> > ### Author Rebuttal · Reviewer_tXTj · 2026-04-03
> >
> > Thanks to the authors for the thorough rebuttal. My concerns are fully resolved.

---

> > > ### Author Response · Authors · 2026-04-03
> > >
> > > Thank you for your feedback and time! We are glad that our rebuttal has fully resolved your concerns.

---

### Decision · Program_Chairs · 2026-04-30

**Decision:**

Accept (regular)

**Comment:**

The paper introduces "Proximal Decoding," an inference-time technique designed to prevent large language models (trained on mixed or risky data) from verbatim regurgitation of their (potentially copyrighted) training test. Therefore, it  runs a "safe" language model (trained exclusively on open data) in parallel with the risky model during generation. The method constrains the output token distribution of the risky model to stay within a specified KL-divergence radius of the safe model's distribution.

The reviewers positively highlighted that the paper addresses a timely and important topic with strong theory and practical contribution. Especially the release of Comma 1.7B is considered of high value for the community.

The rebuttal addressed the concerns that were raised including the runtime (where it was shown that the overhead is not too large), the scope of the guarantee (where the token-level guarantee serves as a good proxy to the to the sequence-level), the benefits from stronger models (where experiments with Comma 7B highlighted that improvements can be brought from improvements in the safe models).

The concerns that remained after the rebuttal are mainly about the framing. In case of acceptance, the framing of the paper should be adapted to strongly highlight that this is not an absolute guarantee but rather a guarantee with respect to some safe model. So the todo would be to significantly tone down the framing/title/claims around “provably reducing copyright risk”. Additionally, the discussion should include the focus on the compute costs that might rise with larger safe models (discussing the trade-off between risk reduction and overhead). Finally, either the empirical evaluation should be broadened to include the examples of code and newspaper articles etc. that were added during the rebuttal and extend on them. Alternatively, the claims could also be narrowed down further.